# Sequential MCR via Staudinger/Aza-Wittig versus Cycloaddition Reaction to Access Diversely Functionalized 1-Amino-1*H*-Imidazole-2(3*H*)-Thiones

**DOI:** 10.3390/molecules24203785

**Published:** 2019-10-21

**Authors:** Cecilia Ciccolini, Giacomo Mari, Gianfranco Favi, Fabio Mantellini, Lucia De Crescentini, Stefania Santeusanio

**Affiliations:** Department of Biomolecular Sciences, Section of Chemistry and Pharmaceutical Technologies, University of Urbino “Carlo Bo”, Via I Maggetti 24, 61029 Urbino (PU), Italy; c.ciccolini@campus.uniurb.it (C.C.); giacomo.mari@uniurb.it (G.M.); fabio.mantellini@uniurb.it (F.M.); lucia.decrescentini@uniurb.it (L.D.C.)

**Keywords:** multicomponent reaction, α-halohydrazones, Staudinger reaction, aza-Wittig, 1*H*-imidazole-2(3*H*)-thione, 2*H*-imidazo[2,1-*b*][1,3,4]thiadiazine

## Abstract

A multicomponent reaction (MCR) strategy, alternative to the known cycloaddition reaction, towards variously substituted 1-amino-1*H*-imidazole-2(3*H*)-thione derivatives has been successfully developed. The novel approach involves α-halohydrazones whose azidation process followed by tandem Staudinger/aza-Wittig reaction with CS_2_ in a sequential MCR regioselectively leads to the target compounds avoiding the formation of the regioisomer iminothiazoline heterocycle. The approach can be applied to a range of differently substituted α-halohydrazones bearing also electron-withdrawing groups confirming the wide scope and the substituent tolerance of the process for the synthesis of the target compounds. Interestingly, the concurrent presence of reactive functionalities in the scaffolds so obtained ensures post-modifications in view of *N*-bridgeheaded heterobicyclic structures.

## 1. Introduction

Imidazoles belong to an important class of heterocyclic compounds that play a crucial role in various biochemical processes [1]. A lot of imidazole-based molecules have been shown bioactivities, [2] such as antifungal, antiinflammatory, antihystamine, antihelmintic, analgesic, antineoplastic, antihypertensive activity [3,4,5,6,7].

Among imidazole derivatives, imidazole-2-thiones have been associated to a special class of biologically relevant thiourea derivatives [8] endowed with antithyroid [9], antiproliferative [10], matrix metalloproteinases (MPP) inhibitory [11] properties and can be used as building blocks for the synthesis of *N*-aminoimidazole with antiretroviral activity [12].

To date, the most widespread method used for the synthesis of *N*-substituted 1-amino-1*H*-imidazol-2(3*H*)-thiones can be referred to the Schantl’s protocol, which consists of reacting α-haloketones with potassium thiocyanate and monosubstituted arylhydrazines in weak acidic medium (Scheme 1) [13,14,15,16,17,18,19]. This multistep reaction is considered to proceed via the formation of conjugated azoalkenes, derived from α-thiocyanatohydrazones **D** (Scheme 2) and dipolarophile isothiocyanic acid intermediate that in turn undergo a [3+2] cycloaddition reaction providing substituted 1-arylamino-1*H*-imidazole-2(3*H*)-thione **I** scaffolds [20,21].

Even if this method appears robust, it seems to suffer of some limitations in terms of insertion of electron-withdrawing groups placed on the α-halohydrazone precursors of conjugated azoalkene intermediates. In this regard, for our research purposes, we tried to apply the Schantl’s method reacting 2-chloro-*N*,*N*-dimethyl-3-oxobutanamide (**A**), potassiun thiocyanate (**B**) and *tert*-butyl hydrazinecarboxylate (**C**) in acetic acid to obtain the corresponding *N*-substituted 1-amino-1*H*-imidazole-2(3*H*)-thione derivative **I** but without success. As shown in Scheme 2, instead of the cycloaddition, a 5-exo-dig cyclization reaction leading to 2-iminothiazole **II** took place. This evidence is in agreement with the result obtained by Lagoja and coworkers where a pathway involving the key α-thiocyanatohydrazone intermediate **D** is invoked [12].

The structure of the iminothiazoline **II** was confirmed by comparison of the spectral data of the same compound obtained by means of a different procedure previously described by some of us that foresees the conjugated hydrothiocyanation of the pertinent conjugated azoalkene in acidic medium followed by intramolecular cyclization [22].

Inspired by our previous experience [23], and in order to perform a complete regioselective-oriented method for the desired 1-amino-1*H*-imidazole-2(3*H*)-thiones **I**, we have planned a different strategy that avoids the use of bidentate-nucleophilic reagents such as the potassium thiocyanate. In the construction of **I**, three strategic disconnections between the N1-C2, C2-N3 and N3-C4 were hypothesized (Scheme 3).

We reasoned that the azidation process of the pertinent α-halohydrazone derivative followed by tandem Staudinger/aza-Wittig reaction with CS_2_ could have been a successful route [24,25].

## 2. Results and Discussion

To validate our hypothesis we began to explore the process step by step. Thus, α-chlorohydrazone derivative **1a** [26,27,28,29,30] (2.0 mmol) dissolved in THF (9.0 mL) subjected to α-azidation using an ice-cooled aqueous solution of NaN_3_ [31] (2.0 mmol/1.0 mL) under magnetic stirring at room temperature. After the evaporation of the solvent and an appropriate extraction, the α-azidohydrazone derivative **2a** was obtained in 70% yield. In the next step, the addition of a stoichiometric amount of PPh_3_ to **2a** (1.0 mmol) dissolved in CH_2_Cl_2_ (5.0 mL) furnished the iminophosphorane derivative **3a** by precipitation from the reaction medium (66%). Then, **3a** (0.65 mmol) was dissolved in 5.0 mL of THF/MeOH mixture (4:1) and treated with an excess of CS_2_ at reflux to afford, after column chromatography purification, the corresponding *N*-substituted 1-amino-2,3-dihydro-1*H*-imidazole-2-thione derivative **5a** (53%) arising from intramolecular cyclization of the α-isothiocyanate hydrazone intermediate **4a** (Scheme 4).

Motivated by this result, we aimed to develop a one-pot sequential multicomponent reaction (MCR) [32,33,34,35] as alternative method for regioselective synthesis of a new series of imidazole-2-thione-containing structures as suitable precursors for drug-like compounds [36].

Hence, our new approach to *N*-substituted 1-amino-1*H*-imidazole-2(3*H*)-thiones **5a**–**k** (53%–85%) is depicted in Scheme 5. The whole process that permits the formation of the desired heterocycle can be easily checked by the complete disappearance of the pertinent α-azidohydrazone derivative and by the observation of Ph_3_P=S as byproduct (thin-layer chromatography (TLC) check, see Experimental Section). It is to be noted that for **5a**, the efficiency of the reaction benefits by this latter protocol increasing the overall yield from 25% (obtained employing the step-by-step procedure) to 79% (Table 1). Moreover, the implemented strategy broadens the substitution patterns at the amino-N1 and at C4 of the heterocycle skeleton with electron-withdrawing groups (**5a**–**e**) and tolerates the aromatic (amino-N1) and aliphatic (C4) groups, as for **5j** [15,17,18] (Table 1).

These results not only lie in the wide scenario of the heterocyclic scaffolds obtainable through tandem Staudinger/aza-Wittig sequence [24,25,37,38,39,40,41], but the concurrent presence of reactive functionalities in the target compounds **5a**–**k** ensures post-modifications in view of heterobicyclic structures. In fact, the tautomerism thionoamide/thioloimide permits the introduction of a further element of diversity at the sulfur atom producing imidazole derivatives suitable to be combined with the useful 1-amino-Boc protected group [42] directly installed by this approach, as for **5a**, **5c**–**f, 5k**. Thus, as an example, **5c**,**d**,**f** (1.0 mmol) solved in acetone (10.0 mL), were reacted with 2-bromo-1-phenylethanone (**6a**) (1.0 mmol), 1-chloropropan-2-one) (**6b**) (1.0 mmol), and ethyl 2-bromoacetate (**6c**) (1.0 mmol), respectively, in the presence of K_2_CO_3_ (1.0 mmol). After the removal of solvent followed by extraction, the corresponding α-(imidazol-2-ylthio) carbonyl compounds **7a**–**c** were obtained as solid after column chromatography purification (84%–93%) (Scheme 6). The subsequent cleavage of the Boc-protecting group under homogeneous [43] or heterogeneous acidic conditions [44] was able to produce free amino function available to interact with the carbonyl appendage in 2-position of the ring, affording new 2*H*-imidazo [2,1-*b*][1,3,4]thiadiazine derivatives **8a**,**b** by condensation or 2*H*-imidazo[2,1-b][1,3,4]thiadiazinone derivative **8c** by acylic nucleophilic substitution process (Scheme 6, Table 2).

It is worthwhile to note that the proposed synthetic pathway can offer an alternative method for obtaining 2*H*-imidazo[2,1-*b*][1,3,4]thiadiazine derivatives **8** with respect to the ring transformation of α-(oxazol-2-ylthio) ketones **9** on treatment with hydrazine hydrate **10** [45], together with the possibility of wide diversification of the substituents at the different positions of the *N*-bridgeheaded heterobicyclic structures. As depicted in Scheme 7, a different disconnection for the assembly of the 2*H*-imidazo[2,1-*b*][1,3,4]thiadiazine scaffold can be envisaged.

## 3. Experimental Section

### 3.1. General

All the commercially available reagents and solvents were used without further purification. α-Halohydrazones **1a**–**k** were synthesized by known procedures [26,27,28,29,30]. Chromatographic purification of compounds was carried out on silica gel (60–200 μm). Thin-layer chromatography (TLC) analysis was performed on pre-loaded (0.25 mm) glass supported silica gel plates (Silica gel 60, F254, Merck; Darmstadt, Germany); compounds were visualized by exposure to UV light. Melting points (Mp) were determined in open capillary tubes and are uncorrected.

All ^1^H NMR and ^13^C NMR spectra were recorded at 400 and 100 MHz, respectively at 25 °C on a Bruker Ultrashield 400 spectrometer (Bruker, Billerica, MA, USA). Proton and carbon spectra were referenced internally to residual solvent signals as follows: δ = 2.50 ppm for proton (middle peak) and δ = 39.50 ppm for carbon (middle peak) in DMSO-*d*_6_ and δ = 7.27 ppm for proton and δ = 77.00 ppm for carbon (middle peak) in CDCl_3_. The following abbreviations are used to describe peak patterns where appropriate: s = singlet, d = doublet, t = triplet q = quartet, m = multiplet and br = broad signal. All coupling constants (*J*) are given in Hz. Copies of ^1^H-NMR and ^13^C-NMR spectra of compounds **II**, **2a**, **3a**, **5a**–**k**, **7a**–**c**, and **8a**–**c** are in Appendix A. FT-IR spectra were measured as Nujol mulls using a Nicolet Impact 400 (Thermo Scientific, Madison, WI, USA). Mass spectra were obtained by ESI-MS analyses performed on Thermo Scientific LCQ Fleet Ion Trap LC/MS and Xcalibur data System. High-resolution mass spectra (HRMS) were determined with ESI resource on a Waters Micromass QTOF instrument (Waters, Milford, MA, USA). Elemental analyses were within ±0.4 of the theoretical values (C, H, N).

### 3.2. Step-By-Step Synthetic Method for **5a**

#### 3.2.1. Synthesis of *tert*-butyl 2-(3-azido-4-(dimethylamino)-4-oxobutan-2-ylidene)hydrazinecarboxylate (**2a**)

To the α-halohydrazone **1a** (555.5 mg, 2.0 mmol) solved in THF (9.0 mL), an ice-cooled aqueous solution (1.0 mL, T = 4 °C) of NaN_3_ (2.0 mmol, 130.02 mg) was added. The reaction mixture was stirred at room temperature until the disappearance of the starting **1a** (TLC check). THF was removed under reduced pressure and the residue was diluted with water and extracted with CH_2_Cl_2_ (3 × 15.0 mL). The combined organic layers were dried over anhydrous NaSO_4_ and concentrated under reduced pressure. The crude reaction was purified by crystallization from Et_2_O affording the α-azido derivative **2a**. Yield 70.0% (398.0 mg) as a white solid; Mp 120–124 °C (dec); ^1^H-NMR, 400 MHz, DMSO-*d_6_*) δ 1.44 (s, 9H, OBu*^t^*), 1.84 (s, 3H, CH_3_), 2.86 (s, 3H, NCH_3_), 2.92 (s, 3H, NCH_3_), 4.99 (s, 1H, CH), 9.82 (br s, 1H, NH, D_2_O exch.); ^13^C-NMR (100 MHz, DMSO-*d_6_*) δ 13.8, 28.0, 35.5, 36.6, 64.6, 79.5, 146.2, 152.9, 166.5; IR (Nujol, ν, cm^−1^): 3239, 3150, 2982, 2172, 2098, 1706, 1686, 1664; MS *m*/*z* (ESI): 285.07 (M + H)^+^; anal. calcd. for C_11_H_20_N_6_O_3_ (284.31): 46.47; H, 7.09; N, 29.56; found: C, 46.36; H, 7.15; N, 29.65.

#### 3.2.2. Synthesis of *tert*-butyl 2-(4-(dimethylamino)-4-oxo-3-((triphenylphosphoranylidene)amino)butan-2-ylidene)hydrazinecarboxylate (**3a**)

1.0 Mmol of **2a** (284.31 mg) was solved in CH_2_Cl_2_ (5.0 mL). The reaction flask was then immersed in an ice bath (T = 0 °C), and a cooled solution of PPh_3_ (262.3 mg, 1.0 mmol) in CH_2_Cl_2_ (1.0 mL) was added dropwise. The reaction was brought back to room temperature and stirred until the disappearance of organic azide **2a** (monitored by TLC). The formation of phosphazene **3a** was accompanied by the development of N_2_. After partial removal of the solvent under reduced pressure, **3a** was isolated by precipitation from a solution of CH_2_Cl_2_/EtOAc as white powder; yield 66% (342.3 mg); Mp 127–131 °C (dec.); ^1^H-NMR (400 MHz, DMSO-*d_6_*) δ 1.42 (s, 9H, OBu*^t^*), 1.81 (s, 3H, CH_3_), 2.59 (s, 3H, NCH_3_), 2.73 (s, 3H, NCH_3_), 4.62 (t, *J*_H-P_ = 9.2 Hz, 1H, CH), 7.57–7.90 (m, 15H, Ar), 9.64 (s, 1H, NH, D_2_O exch.) ppm; ^13^C-NMR (100 MHz, DMSO-*d_6_*) δ 13.0, 28.0, 35.6, 36.1, 59.3, 79.5, 120.9 (^1^*J*_C-P_ = 102.0 Hz), 129.7 (^2^*J*_C-P_ = 14.0 Hz), 133.7 (^3^*J*_C-P_ = 11.0 Hz), 133.8 (^3^*J-*_CP_ = 12.0 Hz), 134.9 (^4^*J*_C-P_ = 2.0 Hz), 150.7, 167.0 ppm; IR (Nujol, ν, cm^−1^): 3543, 3377, 3211, 1722, 1664; MS *m*/*z* (ESI): 519.31 (M + H)^+^; anal. calcd. for C_29_H_35_N_4_O_3_P (518.59): C, 67.17; H, 6.80; N, 10.80; found: C, 67.31; H, 6.86; N, 10.72.

#### 3.2.3. Synthesis of *tert*-butyl (4-(dimethylcarbamoyl)-5-methyl-2-thioxo-2,3-dihydro-1H-imidazol-1-yl)carbamate (**5a**)

0.65 Mmol of **3a** (337.0 mg,) was solved in a mixture of THF:MeOH (4:1, 5.0 mL) heating. Then, 0.5 mL of CS_2_ was added and the reaction was refluxed. The end of the reaction was defined (4.0 h) by the disappearance of **3a** together with the formation of Ph_3_P=S as byproduct (monitored by TLC). After removal of the reaction solvents under reduced pressure, a first crop of **5a** was obtained as white powder from a solution of THF/light petroleum ether. A further amount was be gained by column chromatography eluting with CH_2_Cl_2_/EtOAc mixtures. White powder from THF/light petroleum ether; yield 53% (103.4 mg); Mp 172–173 °C (dec.); ^1^H-NMR (400 MHz, DMSO-*d_6_*,) δ 1.32 and 1.45 (2 s, 9H, OBu*^t^*), 1.99 (s, 3H, CH_3_), 2.94 [s, 6H, N(CH_3_)_2_], 9.69 and 10.15 (2 br s, 1H, NH, D_2_O exch.), 12.50 (br s, 1H, NH, D_2_O exch.) ppm; ^13^C-NMR (100 MHz, DMSO-*d_6_*) δ 8.9, 27.6, 27.8, 35.9, 80.8, 116.2, 128.2, 153.8, 160.2, 162.9 ppm; IR (Nujol, ν, cm^−1^): 3188, 3115, 1741, 1645, 1607; MS *m*/*z* (ESI): 301.15 (M + H)^+^; calcd. for C_12_H_20_N_4_O_3_S (300.38): C, 47.98; H, 6.71; N, 18.65; found: C, 48.11; H, 6.63; N, 18.57. The partition of some signals here, as well as in the following cases, is due to the N1-amide rotameric effect [46].

### 3.3. Typical MCR Procedure for the Synthesis of N-Substituted 1-Amino-1H-Imidazole-2(3H)-Thione Derivatives **5a**–**k**

To a round flask equipped with a magnetic stirring bar containing ice-cooled solution of NaN_3_ (1.0 mmol, 65.01 mg) dissolved in 0.5 mL of H_2_O, the corresponding α-halohydrazone **1a**–**k** (1.0 mmol) dissolved in THF (4.5 mL) was added. The mixture was stirred at room temperature until the disappearance of **1** (monitored by TLC). Upon completion, Na_2_SO_4_ (0.5 g)_,_ a solution of PPh_3_ (1.1 mmol, 288.5 mg) in THF (1.0 mL) and CS_2_ (1.0 mL) were added in sequence, and the mixture was refluxed for the appropriate reaction time (3.0–20.0 h). The formation of the final products **5a**–**k** was revealed by the complete disappearance of the spot corresponding to the α-azidohydrazone **2a**–**k** as well as the detection of the byproduct Ph_3_P=S. The Na_2_SO_4_ was filtered in vacuo and washed with THF (10.0 mL). The filtrate was concentrated under reduced pressure and the residue was purified by crystallization and/or by chromatography eluting with cyclohexane:EtOAc or CH_2_Cl_2_:EtOAc mixtures. The resulting products **5a**–**k** were isolated by crystallization from the specific solvents (see below). According to this procedure, **5a** was obtained in 79% (237.3 mg).

*N,N,5-trimethyl-1-(3-phenylureido)-2-thioxo-2,3-dihydro-1H-imidazole-4-carboxamide* (**5b**): Yield 53% (169.3 mg), pink powder from CH_2_Cl_2_/Et_2_O; Mp 247–248 °C (dec.); ^1^H-NMR (400 MHz, DMSO-d_6_) δ 2.06 (s, 3H, CH_3_), 2.97 [s, 6H, N(CH_3_)_2_], 7.01 (t, *J* = 8.0 Hz, 1H, Ar), 7.29 (t, *J* = 8.0 Hz, 2H, Ar), 7.46 (d, *J* = 8.0 Hz, 2H, Ar), 9.00 (s, 1H, NH, D_2_O exch.), 9.33 (br s, 1H, NH, D_2_O exch.), 12.56 (s, 1H, NH, D_2_O exch.) ppm; ^13^C-NMR (100 MHz, DMSO-d_6_) δ 9.3, 36.8, 116.1, 118.3, 122.3, 128.7, 129.0, 139.1, 153.6, 160.3, 162.6 ppm; IR (Nujol, ν, cm^−1^): 3323, 3248, 3195, 3136, 1713, 1638, 1605; MS *m*/*z* (ESI): 320.40 (M + H)^+^; calcd. for; C_14_H_17_N_5_O_2_S (319.38): C, 52.65; H, 5.37; N, 21.93; calcd. for; C_14_H_17_N_5_O_2_S (319.38): C, 52.65; H, 5.37; N, 21.93; found: C, 52.79; H, 5.44; N, 21.84.

*tert-Butyl (4-(diethylcarbamoyl)-5-methyl-2-thioxo-2,3-dihydro-1H-imidazol-1-yl)carbamate (***5c***):* Yield 72% (236.3 mg), white powder from EtOAc/THF/light petroleum ether; Mp 168–169 °C (dec.); ^1^H-NMR (400 MHz, DMSO-d_6_) δ 1.06–1.10 (m, 6H, 2xNCH_2_CH_3_), 1.32 and 1.45 (2s, 9H, OBu^t^), 1.94 and 1.97 (2s, 3H, CH_3_), 3.26–3.37 (m, 4H, 2xNCH_2_CH_3_), 9.68 and 10.07 (2 br s, 1H, NH, D_2_O exch.), 12.49 (br s, 1H, NH, D_2_O exch.) ppm; ^13^C-NMR (100 MHz, DMSO-d_6_) δ 8.6, 13.4, 27.5, 27.8, 34.8, 80.6, 117.0, 126.6, 153.8, 159.8, 162.7 ppm; IR (Nujol, ν, cm^−1^): 3169, 3120, 1748, 1642, 1634; MS *m*/*z* (ESI): 329.23 (M + H)^+^;calcd. for C_14_H_24_N_4_O_3_S (328.16): C, 51.20; H, 7.37; N, 17.06; found: C, 51.09; H, 7.42; N, 16.95.

*tert-Butyl (5-methyl-2-thioxo-2,3-dihydro-1H-imidazol-1-yl)carbamate* (**5d**): Yield 69% (158.1 mg), white powder from EtOAc/THF/light petroleum ether; Mp 168–169 °C (dec.); ^1^H-NMR (400 MHz, DMSO-d_6_) δ 1.32 and 1.45 (2s, 9H, OBu^t^), 1.93 (s, 3H, CH_3_), 6.60 (s, 1H, CH), 9.51 and 9.94 (2 br s, 1H, NH, D_2_O exch.), 11.97 (br s, 1H, NH, D_2_O exch.) ppm; ^13^C-NMR (100 MHz, DMSO-d_6_) δ 8.9, 27.9, 80.5, 108.9, 126.9, 153.9, 162.4 ppm; IR (Nujol, ν, cm^−1^): 3271, 3144, 3098, 1744, 1732, 1640; MS *m*/*z* (ESI): 229.96 (M + H)^+^; calcd. for C_9_H_15_N_3_O_2_S (229.09): C, 47.14; H, 6.59; N, 18.33; found: C, 47.01; H, 6.65; N, 18.41.

*tert-Butyl (4-carbamoyl-5-methyl-2-thioxo-2,3-dihydro-1H-imidazol-1-yl)carbamate* (**5e**): Yield 58% (157.8 mg), white powder from CH_2_Cl_2/_light petroleum ether; Mp 270 °C (dec.); ^1^H-NMR (400 MHz, DMSO-d_6_) δ 1.32 and 1.45 (2s, 9H, OBu^t^), 2.23 and 2.26 (2s, 3H, CH_3_), 7.23 and 7.53 (2 br s, 2H, NH_2_, D_2_O exch.), 9.71 and 10.17 (2s, 1H, NH, D_2_O exch.), 12.42 (s, 1H, NH, D_2_O exch.) ppm; ^13^C NMR (100 MHz, DMSO-d_6_) δ 8.9, 10.1, 27.6, 27.8, 80.9, 115.8, 133.1, 153.7, 159.6, 162.9 ppm; IR (Nujol, ν, cm^−1^): 3395, 3354, 3182, 3137, 1754, 1717, 1676, 1594; MS *m*/*z* (ESI): 273.04 (M + H)^+^; calcd. for C_10_H_16_N_4_O_3_S (272.09): C, 44.10; H, 5.92; N, 20.57; found: C, 44.23; H, 5.96; N, 20.45.

*tert-Butyl (5-methyl-4-(phenylcarbamoyl)-2-thioxo-2,3-dihydro-1H-imidazol-1-yl)carbamate* (**5f**): Yield 67% (233.2 mg), white powder from EtOAc; Mp 170–171 °C (dec.); ^1^H-NMR (400 MHz, DMSO-d_6_) δ 1.34 and 1.46 (2s, 9H, OBu^t^), 2.28 (s, 3H, CH_3_), 7.11 (t, *J* = 8.0 Hz, 1H, Ar), 7.35 (t, *J* = 8.0 Hz, 2H, Ar), 7.65 (d, *J* = 8.0 Hz, 2H, Ar), 9.68 (s, 1H, NH, D_2_O exch.), 10.28 (s, 1H, NH, D_2_O exch.), 12.69 (s, 1H, NH, D_2_O exch.) ppm; ^13^C-NMR (100 MHz, DMSO-d_6_) δ 9.2, 27.8, 81.0, 116.0, 119.7, 123.8, 128.8, 133.9, 138.4, 153.7, 156.3, 163.2 ppm; IR (Nujol, ν, cm^−1^): 3375, 3243, 3066, 1752, 1659, 1630, 1598, 1545; MS *m*/*z* (ESI): 349.22 (M + H)^+^; calcd. for C_16_H_20_N_4_O_3_S (348.13): C, 55.16; H, 5.79; N, 16.08; found: C, 55.01; H, 5.72; N, 16.16.

*1-(5-Methyl-2-thioxo-2,3-dihydro-1H-imidazol-1-yl)-3-phenylurea* (**5g**): Yield 82% (203.4 mg), white powder from THF/EtOAc; Mp 245–248 °C (dec.); ^1^H-NMR (400 MHz, DMSO-d_6_) δ 2.01 (s, 3H, CH_3_), 6.64 (s, 1H, CH), 6.99 (t, *J* = 8.0 Hz, 1H, Ar), 7.28 (t, *J* = 8.0 Hz, 2H, Ar), 7.46 (d, *J* = 8.0 Hz, 2H, Ar), 8.91 (s, 1H, NH, D_2_O exch.), 9.25 (s, 1H, NH, D_2_O exch.), 12.05 (s, 1H, NH, D_2_O exch.) ppm; ^13^C-NMR (100 MHz, DMSO-d_6_) δ 9.1, 108.8, 118.3, 122.3, 127.6, 128.8, 139,1, 153.8, 161.9 ppm; IR (Nujol, ν, cm^−1^): 3305, 3154, 3119, 3097, 1714, 1681, 1637, 1602; MS *m*/*z* (ESI): 249.07 (M + H)^+^; calcd. for C_11_H_12_N_4_OS (248.07): C, 53.21; H, 4.87; N, 22.56; found: C, 53.08; H, 4.94; N, 22.65.

*N-(5-methyl-2-thioxoimidazolidin-1-yl)benzamide* (**5h**): Yield 59% (137.6 mg) white powder from MeOH; Mp 240–242 °C (dec.); ^1^H-NMR (400 MHz, DMSO-d_6_) δ 1.97 (s, 3H, CH_3_), 6.72 (s, 1H, CH), 7.56 (t, *J* = 8.0 Hz, 2H, Ar), 7.65 (t, *J* = 8.0 Hz, 1H, Ar), 7.99 (d, *J* = 8.0 Hz, 2H, Ar), 11.44 (s, 1H, NH, D_2_O exch.), 12.15 (s, 1H, NH, D_2_O exch.) ppm; ^13^C-NMR (100 MHz, DMSO-d_6_) δ 8.9, 109.3, 127.0, 127.7, 128.6, 131.5, 132.5, 162.0, 165.4 ppm; IR (Nujol, ν, cm^−1^): 3168, 3106, 1666, 1631; MS *m*/*z* (ESI): 234.04 (M + H)^+^; calcd. for C_11_H_11_N_3_OS (233.29): C, 56.63; H, 4.75; N, 18.01; found: C, 56.76; H, 4.82; N, 17.89.

*1-(4,5-Dimethyl-2.thioxo-2,3-dihydro-1H-imidazol-1-yl)-3-phenylurea* (**5i**): Yield 85% (223.0 mg), white powder from THF/Et_2_O; Mp 245–250 °C (dec.); ^1^H-NMR (400 MHz, DMSO-d_6_) δ 1.94 (s, 3H, CH_3_), 1.99 (s, 3H, CH_3_), 6.99 (t, *J* = 8.0 Hz, 2H, Ar), 7.28 (t, *J* = 8.0 Hz, 1H, Ar), 7.46 (d, *J* = 8.0 Hz, 2H, Ar), 8.89 (s, 1H, NH, D_2_O exch.), 9.19 (s, 1H, NH, D_2_O exch.), 12.00 (s, 1H, NH, D_2_O exch.) ppm; ^13^C-NMR (100 MHz, DMSO-d_6_) δ 7.8, 8.9, 116.6, 118.3, 122.2, 122.5, 128.7, 139.1, 153.9, 160.7 ppm; IR (Nujol, ν, cm^−1^): 3271, 3172, 3095, 1719, 1691, 1665, 1603; MS *m*/*z* (ESI): 263.11 (M + H)^+^; calcd. for C_12_H_14_N_4_OS (262.33): C, 54.94; H, 5.38; N, 21.36; found: C, 54.87; H, 5.46; N, 21.23.

*4,5-Dimethyl-1-[(4-nitrophenyl)amino]-1H-imidazole-2(3H)-thione* (**5j**): Yield 84% (222.0 mg), beige powder from THF/EtOAc/Et_2_O; Mp 279–282 °C (dec.); ^1^H-NMR (400 MHz, DMSO-d_6_) δ 1.90 (s, 3H, CH_3_), 2.03 (s, 3H, CH_3_), 6.59 (d, *J* = 8.0 Hz, 2H, Ar), 8.10 (d, *J* = 8.0 Hz, 2H, Ar), 10.09 (s, 1H, NH, D_2_O exch.), 12.19 (s, 1H, NH, D_2_O exch.) ppm; ^13^C-NMR (100 MHz, DMSO-d_6_) δ 7.6, 9.0, 111.3, 117.8, 121.7, 125.8, 139.3, 153.0, 160.8 ppm; IR (Nujol, ν, cm^−1^): 3199, 3094, 1673, 1594; HRMS *m*/*z* calcd. for [M + H]^+^ C_11_H_13_N_4_O_2_S 265.0759; found 265.0774.

*tert-Butyl (5-phenyl-2-thioxo-2,3-dihydro-1H-imidazol-1-yl)carbamate* (**5k**): Yield 66% (192.3 mg), light yellow powder from THF/EtOAc/light petroleum ether; Mp 172–174 °C (dec.); ^1^H-NMR (400 MHz, DMSO-d_6_) δ 1.17 and 1.39 (2s, 9H, OBu^t^), 7.18 (s, 1H, CH), 7.35–7.49 (m, 5H, Ar), 9.79 and 10.12 (2s, 1H, NH, D_2_O exch.), 12.51 (br s, 1H, NH, D_2_O exch.) ppm; ^13^C-NMR (100 MHz, DMSO-d_6_) δ 27.5, 27.9, 80.2, 80.5, 110.7, 110.8, 126.9, 127.0, 127.7, 127.9, 128.1, 128.5, 130.6, 130.8, 153.2, 153.9, 162.0, 164.2 ppm; IR (Nujol, ν, cm^−1^): 3275, 3120, 3093, 1726, 1618, 1600; MS *m*/*z* (ESI): 292.18 (M + H)^+^; calcd. for C_14_H_17_N_3_O_2_S (291.37); C, 57.71; H, 5.88; N, 14.42; found: C, 57.83; H, 5.82; N, 14.37.

### 3.4. General Procedure for the Synthesis of α-(Imidazol-2-Ylthio) Carbonyl Compounds **7a**–**c**.

To a suspension of the *N*-Boc-protected 1-amino-1*H*-imidazole-2(3*H*)-thione derivatives **5c,d**,**f** (1.0 mmol) and K_2_CO_3_ (1.0 mmol, 138 mg) in 10.0 mL of acetone, the corresponding α-halocarbonyl derivative **6a**–**c** (1.0 mmol) was added. The reaction mixture was kept under magnetic stirring at room temperature. Upon completion (monitored by TLC) the solvent was removed, and the crude reaction mixture was quenched to neutrality with a solution of HCl 1N and extracted with EtOAc (30.0 mL). The organic layer was washed with brine and dried over anhydrous Na_2_SO_4_. The solvent was removed in vacuum and the crude extract was purified by crystallization or by column chromatography eluting with cyclohexane:ethyl acetate mixtures to furnish **7a**–**c** derivatives in good yields (84%–93%).

*tert-Butyl (4-(diethylcarbamoyl)-5-methyl-2-((2-oxo-2-phenylethyl)thio)-1H-imidazol-1-yl)carbamate* (**7a**): Yield 84% (375.1 mg); white solid from Et_2_O; Mp 123–126 °C; ^1^H-NMR (400 MHz, CDCl_3_) δ 1.14–1.23 (m, 6H, 2xNCH_2_CH_3_), 1.48 (s, 9H, OBu^t^), 1.98 (s, 3H, CH_3_), 3.37–3.64 (m, 4H, 2xNCH_2_CH_3_), 4.50 (br s, 1H, SCH_a_H_b_), 4.69 (br s, 1H, SCH_a_H_b_), 7.46 (t, *J* = 8.0 Hz, 2H, Ar), 7.58 (t, *J* = 8.0 Hz, 1H, Ar), 7.97 (d, *J* = 8.0 Hz, 2H, Ar), 9.45 (br s, 1H, NH, D_2_O exch.) ppm; ^13^C-NMR (100 MHz, CDCl_3_) δ 8.7, 12.9, 14.5, 28.0, 40.6, 41.6, 43.4, 82.5, 128.4, 128.5, 128.7, 130.9, 133.7, 135.3, 135.4, 139.9, 153.8, 164.5, 193.7 ppm; IR (Nujol, ν, cm^−1^): 3114, 3059, 1741, 1726, 1700, 1681, 1597, 1584; MS *m*/*z* (ESI): 447.35 (M + H)^+^; calcd. for C_22_H_30_N_4_O_4_S (446.56): C, 59.17; H, 6.77; N, 12.55; found: C, 59.02; H, 6.84; N, 11.67.

*tert-Butyl (5-methyl-2-((2-oxopropyl)thio)-1H-imidazol-1-yl)carbamate* (**7b**): Yield 93% (265.4 mg); ocher solid from EtOAc/cyclohexane; Mp 103–104 °C; ^1^H-NMR (400 MHz, CDCl_3_) δ 1.50 (s, 9H, OBu^t^), 2.14 (s, 3H, CH_3_), 2.24 (s, 3H, COCH_3_), 3.81 (br s, 1H, SCH_a_H_b_), 3.94 (br s, 1H, SCH_a_H_b_), 6.77 (s, 1H, CH), 8.26 (br s, 1H, NH, D_2_O exch.) ppm; ^13^C-NMR (100 MHz, CDCl_3_) δ 8.9, 28.0, 28.8, 44.8, 82.8, 125.1, 131.5, 139.5, 154.1, 203.5 ppm; IR (Nujol, ν, cm^−1^): 3125, 1725, 1714; MS *m*/*z* (ESI): 286.17 (M + H)^+^; calcd. for C_12_H_19_N_3_O_3_S (285.36): C, 50.51; H, 6.71; N, 14.73; found: C, 50.40; H, 6.78; N, 14.86.

*Ethyl 2-((1-((tert-Butoxycarbonyl)amino)-5-methyl-4-(phenylcarbamoyl)-1H-imidazol-2-yl)thio)acetate* (**7c**): Yield 92% (399.7 mg), white solid from EtOAc/cyclohexane; Mp 134–136 °C; ^1^H-NMR (400 MHz, CDCl_3_) δ 1.27 (t, *J* = 8.0 Hz, 3H, OCH_2_CH_3_), 1.51 (s, 9H, OBu^t^), 2.55 (s, 3H, CH_3_), 3.66 (d, *J* = 16.0 Hz, 1H, SCH_a_H_b_), 3.90 (d, *J* = 16.0 Hz, 1H, SCH_a_H_b_), 4.17–4.24 (m, 2H, OCH_2_CH_3_), 7.10 (t, *J* = 8.0 Hz, 1H, Ar), 7.34 (t, *J* = 8.0 Hz, 2H, Ar), 7.67 (d, *J* = 8.0 Hz, 2H, Ar), 8.18 (br s, 1H, NH, D_2_O exch.), 8.96 (s, 1H, NH, D_2_O exch.) ppm; ^13^C-NMR (100 MHz, CDCl_3_): δ 9.4, 14.0, 28.0, 36.7, 62.7, 83.4, 119.5, 123.8, 128.9, 129.9, 137.3, 138.0, 139.0, 153.6, 160.6, 169.9 ppm; IR (Nujol, ν, cm^−1^): 3315, 3182, 1733, 1647, 1601; MS *m*/*z* (ESI): 435.20 (M + H)^+^; calcd. for C_20_H_26_N_4_O_5_S (434.51): C, 55.28; H, 6.03; N, 12.89; found: C, 55.39; H, 5.97; N, 12.81.

### 3.5. General Procedure for the Synthesis of N-Bridgeheaded Heterobicyclic Derivatives **8a**–**c**.

Derivative **7a,b** (1.0 mmol) was solved in 5.0 mL of a solution of trifluoroacetic acid (TFA) and CH_2_Cl_2_ (1:1). The reaction mixture was left at room temperature until the disappearance of the starting **7a**,**b** (TLC check). Then, the solvent was removed under reduced pressure and the crude reaction mixture was quenched to neutrality with a saturated solution of Na_2_CO_3_ and extracted with EtOAc (20.0 mL × 3). The combined organic layers were washed with brine and dried over anhydrous Na_2_SO_4_. After the removal of the solvent, the crude extract was purified by crystallization or by column chromatography eluting with cyclohexane/ethyl acetate mixtures to furnish **8a**,**b** derivatives. For obtaining **8c**, the best condition found was to treat **7c** (1.0 mmol) with Amberlyst 15H (500 mg) in refluxing dioxane (15.0 mL) for 12.0 h. Upon completion (monitored by TLC) the resin was filtered off in vacuo and washed with THF (20.0 mL). The filtrate was evaporated under reduced pressure and the crude reaction mixture was purified by crystallization.

*N,N-diethyl-6-methyl-3-phenyl-2H-imidazo[2,1-b][1,3,4]thiadiazine-7-carboxamide* (**8a**): Yield 82% (269.3 mg) white powder from EtOAc/cyclohexane; Mp 125–127 °C; ^1^H-NMR (400 MHz, CDCl_3_) δ 1.22 (t, *J* = 8.0 Hz, 6H, 2xNCH_2_CH_3_), 2.56 (s, 3H, CH_3_), 3.53 (br s, 2H, NCH_2_CH_3_), 3.74 (br s, 2H, NCH_2_CH_3_), 3.97 (s, 2H, SCH_2_), 7.50–7.52 (m, 3H, Ar), 7.90 (d, *J* = 8.0 Hz, 2H, Ar) ppm; ^13^C-NMR (100 MHz, CDCl_3_) δ 9.7, 13.0, 14.5, 23.9, 40.3, 43.0, 127.0, 128.9, 129.7, 131.1, 131.3, 133.5, 134.2, 150.8, 164.1 ppm; IR (Nujol, ν, cm^−1^): 1611, 1574, 1562, 1557; MS *m*/*z* (ESI): 329.28 (M + H)^+^; calcd. for C_17_H_20_N_4_OS (328.43): C, 62.17; H, 6.14; N, 17.06; found: C, 62.04; H, 6.19; N, 17.15.

*3,6-Dimethyl-2H-imidazo[2,1**-b][1,3,4]thiadiazine* (**8b**): Yield 65% (108.7 mg); white needles from CHCl_3_/cyclohexane; Mp 57–58 °C; ^1^H-NMR (400 MHz, CDCl_3_) δ 2.26 (s, 3H, CH_3_), 2.32 (s, 3H, CH_3_), 3.42 (s, 2H, SCH_2_), 6.70 (s, 1H, CH), ppm; ^13^C-NMR (100 MHz, CDCl_3_) δ 8.8, 23.5, 26.1, 123.7, 128.6, 130.1, 152.0 ppm; IR (Nujol, ν, cm^−1^): 1640. 1582; MS *m*/*z* (ESI): 168.06 (M + H)^+^; calcd. for C_7_H_9_N_3_S (167.23): C, 50.27; H, 5.42; N, 25.13; found: C, 50.39; H, 5.39 N, 25.06.

*6-Methyl-3-oxo-N-phenyl-3,4-dihydro-2H-imidazo[2,1-**b][1,3,4]thiadiazine-7-carboxamide* (**8c**): Yield 74% (213.3 mg), light yellow powder from EtOAc/light petroleum ether; Mp 229–232 °C; ^1^H-NMR (400 MHz, DMSO-d_6_) δ 2.53 (s, 3H, CH_3_), 3.81 (s, 2H, SCH_2_), 7.04 (t, *J* = 8.0 Hz, 1H, Ar), 7.29 (t, *J* = 8.0 Hz, 2H, Ar), 7.81 (d, *J* = 8.0 Hz, 2H, Ar), 9.80 (s, 1H, NH, D_2_O exch.), 12.27 (br s, 1H, NH, D_2_O exch.) ppm; ^13^C-NMR (100 MHz, DMSO-d_6_): δ 9.2, 29.7, 119.8, 123.0, 128.3, 128.4, 130.6, 132.2, 138.8, 160.8, 164.6 ppm; IR (Nujol, ν, cm^−1^): 1679, 1666, 1595, 1582; MS *m*/*z* (ESI): 288.97 (M + H)^+^; calcd. for C_13_H_12_N_4_O_2_S (288.32): C, 54.15; H, 4.20; N, 19.43; found: C, 54.08; H, 4.27; N, 19.31.

## 4. Conclusions

In conclusion, combining sequential azidation, Staudinger, and aza-Wittig reactions with CS_2_ on α-halohydrazones in a one-pot protocol, variously substituted 1-amino-1*H*-imidazole-2(3*H*)-thiones are directly accessible in good yields and with complete control of regioselectivity. The method is particularly attractive and advantageous for its mild conditions, operational simplicity, and its efficiency as well as its robustness (wide substrate scope and tolerance of various functional groups) and reliability. The concurrent presence of reactive appendages on the obtained scaffolds ensures post-modifications toward *N*-bridgeheaded heterobicyclic structures.

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
