# Peer review of "Sequential MCR via Staudinger/Aza-Wittig versus Cycloaddition Reaction to Access Diversely Functionalized 1-Amino-1H-Imidazole-2(3H)-Thiones"

_molecules, 2019, doi:10.3390/molecules24203785_

Round 1
Reviewer 1 Report
This manuscript describes alterative synthesis of 1-amino-imidazole-thiones. This heterocycle is important biologically active compounds.
Authors established the synthesis of desired imidazole-thiones by MCR reaction. Even EWG substituted substrates, previously difficult ones to apply, are available. Further, conversion of products thiadiazine 8 were successful as an alternative method.
I recommend this manuscript for publication, provided that the following issues are addressed.
The contents of line 78-88 and line 99-105 are duplicated. The latter one will not be necessary. Because procedure is shown in experimental part. Schemes 4 and 5: It is better to show substrate 1 is left side of arrow as a starting material, and reagents like NaN3 are on the arrow.Author Response
Point 1: Lines 99-105 have been deleted and replaced with lines 100-104 highlighted in yellow.
Point 2: Scheme 4 and Scheme 5 have been modified as suggested by the reviewer
Reviewer 2 Report
This manuscript reports the preparation of a series of 1-amino-1H-imidazole-2(3H)-thiones by means of a multicomponent reaction. The work builds on previous literature but does offer an alternative synthetic strategy to this class of compounds. The work is well presented, although in places the English is a bit clunky - it could benefit from minor editing by a native English speaker.
While the work does merit publication in Molecules, it does not meet the requirements of preliminary, but significant, results for publication as a Communication. It should instead be converted to a regular Article. This would also allow the authors to elaborate a bit more on their approach.
Author Response
Point 1: the manuscript has been converted to a regular article
Point 2: spelling and language have been edited by native English speaker as evidenced by the yellow highlighted parts
Reviewer 3 Report
The work is very interesting. In my opinion, this article can be published after correction. I suggest publication as a regular article and not a communication.
Generally, the most schemes should be corrected.
I propose the following changes:
1.Scheme 1. The R1, R2, R3 and X should be defined.
2.Scheme 2. The R1 should be defined. Structure D and E should be deleted.
3.Scheme 3. The hypothesized disconnection should be confirmed by theoretical calculation. It is missing in the text. Therefore, scheme 3 is unnecessary.
4.Scheme 3. Structure 4a should be deleted.
5.Scheme 7 is not clear. In my opinion it should be more discussed in the text.
Author Response
The manuscript has been converted to a regular article
Point 1: Scheme 1 has been modified as suggested
Point 2: Scheme 2 had a typo (R1) and has been corrected. In order to facilitate the reading of the manuscript, we believe that structures D and E are useful to explain the mechanism of formation of the regioisomer 2-iminothiazoline II.
Point 3 and 4: in Scheme 3 the hypothesized disconnection has been supported by similar reactivity within References 20(a) and 21(b), therefore intermediate 4a can be invoked as precursor of 5a
Point 5: Scheme 7 has been improved and its discussion in the text has been implemented (see the yellow highlighted part)